# GSK3β, a Master Kinase in the Regulation of Adult Stem Cell Behavior

**DOI:** 10.3390/cells10020225

**Published:** 2021-01-24

**Authors:** Claire Racaud-Sultan, Nathalie Vergnolle

**Affiliations:** IRSD, INSERM, INRAE, ENVT, Université de Toulouse, UPS, CHU Purpan, Place du Dr Baylac, CEDEX 3, 31024 Toulouse, France; nathalie.vergnolle@inserm.fr

**Keywords:** GSK3b, stem cell, integrin, PAR_2_

## Abstract

In adult stem cells, Glycogen Synthase Kinase 3β (GSK3β) is at the crossroad of signaling pathways controlling survival, proliferation, adhesion and differentiation. The microenvironment plays a key role in the regulation of these cell functions and we have demonstrated that the GSK3β activity is strongly dependent on the engagement of integrins and protease-activated receptors (PARs). Downstream of the integrin α_5_β_1_ or PAR_2_ activation, a molecular complex is organized around the scaffolding proteins RACK1 and β-arrestin-2 respectively, containing the phosphatase PP2A responsible for GSK3β activation. As a consequence, a quiescent stem cell phenotype is established with high capacities to face apoptotic and metabolic stresses. A protective role of GSK3β has been found for hematopoietic and intestinal stem cells. Latters survived to de-adhesion through PAR_2_ activation, whereas formers were protected from cytotoxicity through α_5_β_1_ engagement. However, a prolonged activation of GSK3β promoted a defect in epithelial regeneration and a resistance to chemotherapy of leukemic cells, paving the way to chronic inflammatory diseases and to cancer resurgence, respectively. In both cases, a sexual dimorphism was measured in GSK3β-dependent cellular functions. GSK3β activity is a key marker for inflammatory and cancer diseases allowing adjusted therapy to sex, age and metabolic status of patients.

## 1. Introduction

The Glycogen Synthase Kinase 3β (GSK3β) is an ancestral protein kinase, ubiquitously expressed, which controls many cellular functions [1]. Inhibited in basal conditions, GSK3β is activated under stress and can drive cells to death or to survival depending on its sub-cellular localization and specific partners. GSK3β has been found activated in pathologies such as inflammation and cancer where the deregulation of adult stem cells plays a critical role [2]. Here, we will summarize some key GSK3β-dependent regulations of adult stem cells and their consequences on the behavior of hematopoietic and intestinal stem cells, as examples. We will highlight the importance of adhesion and protease-activated receptors engagement in the activation of signaling pathways conducting to GSK3β activation. Finally, we will discuss the positive and negative aspects of GSK3β activation in pathological conditions and the benefit to consider GSK3β as diagnostic marker and therapeutic target for a precision medicine.

## 2. GSK3β as a Sensor of the Adult Stem Cell Niche

Adult stem cells are in charge of tissue regeneration along the life of organisms. This is the reason why they must be protected from stressors leading to cell death or exhaustion. The microenvironment of adult stem cells, namely “niche”, supports this protective role offering in particular an adhesive anchorage and proper metabolic exchanges [3]. However, aging and exogenous aggressions decrease the niche capacities to control stem cell behavior and thus favor pathogenesis.

Facing stress, adult stem cells could undergo apoptosis, or exhaustion through accelerated differentiation, or senescence, or tumor transformation. As a first step, stem cells must survive and cope with nutrient or oxidative stress. Through its strategic locations in the cell (plasma membrane, endoplasmic reticulum, mitochondria, nucleus), and in response to various signals, GSK3β is able to control death-signaling pathways at the membranes, energetic metabolism and gene transcription [1]. Thus, a resistant quiescent stem cell phenotype is established. It is linked to a strong anchorage to extracellular matrix and supporting cells in the niche, as well as a low energetic metabolism [4,5].

GSK3β acts as a key sensor of cell metabolism and its activation allows energetic supply and anti-oxidant defenses in a low-glucose microenvironment [6]. Indeed, in basal conditions, Insulin receptor-signaling complex inhibits GSK3β to release glycogenesis and Nrf2-dependent anti-oxidant systems. In addition to its role in survival of adult stem cells, active GSK3β prevents ROS-induced differentiation or senescence [7].

As recapitulated in Figure 1, GSK3β plays a central role in the adaptation of adult stem cells to their microenvironment allowing their long-term maintenance during the whole life of the organism.

## 3. GSK3β and Key Functions of Adult Stem Cells

The implication of GSK3β activation in adult stem cell regulation has first been identified for self-renewal function. Indeed, as a serine-threonine kinase, GSK3β is able to phosphorylate β-catenin, targeting it to the proteasome for degradation and blocking its translocation to the nucleus, normally required for stem cell renewal [8]. In the nucleus, β-catenin is a partner of the transcription factor TCF which promotes self-renewal, a proliferative mode maintaining tissue stem cells. Thus the activation of the GSK3β pool associated with β-catenin in the cytoplasm can push stem cells towards a quiescent state.

Another important aspect of GSK3β-dependent regulation of stem cells is the control of cell differentiation. Active GSK3β can influence the balance between energetic/proliferative and differentiation pathways through the activation of mTOR negative regulators [9] and the degradation of transcription factors promoting lineage commitment [10]. As a result, undifferentiated state of tissue progenitors is favored. Here, it is important to note that GSK3β inhibition has a different impact in adult stem cells and their progeny [11,12,13], suggesting that the regulation of GSK3β is tightly dependent on microenvironment specificities.

Migration of adult stem cells is crucial to tissue repair and tumor transformation. Active GSK3β participates to the turnover of focal adhesions built by adhesion receptors (integrins, cadherins), promoting cell migration [14]. This could contribute to cell plasticity allowing transitions between different lineages and epithelial-mesenchymal transition [15].

Advantages conferred by a GSK3β-dependent plasticity could also be critical for stem cell survival. At both plasma membrane and nucleus, active GSK3β promotes a switch in death receptor and nuclear factor-κB (NF-κB) activities, respectively. Under the influence of active GSK3β, a signaling complex (DDX3/cIAP-1) is established downstream of death receptors, promoting cell survival instead of cell death [16]. In addition, GSK3β phosphorylates the p65 subunit, inducing an IκB-independent activation of NF-κB [17]. The GSK3β-dependent activation of NF-κB triggers the transcription of a specific set of genes [18]. Furthermore, NF-κB target genes are under a GSK3β-dependent epigenetic control [19].

All these GSK3β-dependent functions could be critical in stem cells and depend on different pools of GSK3β (Figure 2). It should be noted that certain types of stem cells require a very tight control of GSK3β activity, such as neural stem cells pushed to cell death by autophagy in case of over-activation of GSK3β [20].

## 4. Integrin-Dependent Activation of GSK3β in Leukemia

A number of works have demonstrated the major role of GSK3β in hematopoietic stem cells homeostasis [21]. By regulating both β-catenin and mTOR signaling, GSK3β controls hematopoietic stem cell renewal and differentiation, respectively. As a result, inhibition of GSK3β promotes the expansion of hematopoietic stem cells and their better engraftment [22], whereas activation of GSK3β is associated with differentiation blockade and egress from the hematopoietic niche [13].

GSK3β has been found activated in many cancers, among them leukemia [23]. In leukemia stem cells, GSK3β controls survival, proliferation and differentiation [21]. However, GSK3β inhibition impacts leukemic stem cells survival in an opposite way to their normal counterparts [22]. We have found GSK3β activation in a subset of leukemic cells independent from Akt kinase for their survival, after adhesion onto fibronectin [24,25,26]. Importantly, adhesion of leukemic progenitors to the hematopoietic niche (osteoblasts), has triggered both GSK3β activation and resistance to cytotoxic drugs [24,26]. GSK3β was found associated in a complex with the α_5_ integrin, the scaffolding protein RACK1, and the phosphatase PP2A responsible for its activation through serine 9-dephosphorylation [25].

Adhesion-dependent activation of GSK3β controls the survival of leukemic progenitors through multiple pathways such as the activation of NF-κB independently from IκB variations [24], the resistance to tumor necrosis factor-α [25] and the modulation of Wnt pathway through the up-regulation of secreted Frizzled-related protein-1 [24]. As a result, a very resistant and quiescent phenotype of cancer stem cell is established.

A major observation is that leukemic stem cells (CD34^+^ CD38^-^ CD123^+^, acute myeloid leukemia) surviving through the GSK3β pathway are from female patients [26]. Interestingly, GSK3β-dependent survival has also been measured in leukemic stem cells from male diabetic patients [unpublished results]. Leukemic stem cells from female patients have been characterized by an up-regulation of the expression of RACK1 [26]. Strikingly, normal hematopoietic stem cells from male donors, but not those from females, have been found dependent on the GSK3β pathway to survive [26]. This is a demonstration of the strong capacities of cancer stem cells to hijack plasticity to develop tumors in a physiological niche [27]. However, other acute myeloid leukemia subsets do not depend on GSK3β to survive and are capable to transform their microenvironment in a leukemic niche with benefits for their development [28]. 

The Figure 3 recapitulates data on integrin-dependent GSK3β activation in leukemic stem cells.

## 5. GSK3β in the Colon Crypt: Role and Controls by Proteases and Their Receptors

Is the protective role of GSK3β in hematopoietic cells applicable to other adult stem cells? Pathologies such as inflammatory bowel diseases (IBD) and colorectal cancer (CRC) are characterized by a high activity of GSK3β in epithelial cells and their microenvironment [29,30,31], but the etiology of this over-activity is poorly understood.

In IBD and CRC, epithelial stem cells, located in a cryptic niche, play a critical role since tissue regeneration is impaired either by down-regulation or by deregulation, respectively. Concomitant with their role in epithelial barrier, adhesive molecules are implicated in regeneration, and the remodeling of the adhesive support is critical for the behavior of stem cells [32]. The dual role of the GSK3β-regulated β-catenin both in support of the cadherin-adhesive function and in TCF-transcription activity is critical for the colon stem cell homeostasis [33]. In colorectal cancer cells, active GSK3β associated with mutated APC is not efficient to promote β-catenin degradation, resulting in Wnt signaling deregulation and increased proliferation. In addition, active GSK3β promotes survival and drug resistance [34,35,36] and the degradation of Hath1, transcription factor essential to the differentiation of the secretory lineage [10].

Proteases are expressed in large amounts during inflammation and cancer [37,38,39,40,41]. They can either modify the niche of stem cells through the release of growth factors or the proteolysis of extracellular matrix, or signal through specific receptors, namely protease-activated receptors (PARs) [41]. Our study by immunofluorescence has demonstrated the expression of PAR_2_ and PAR_1_, two members of the PAR family implicated in IBD and CRC, along the human or murine colon crypt and in colon stem cells (Lgr5^+^, Sox9^+^) [42]. Other groups have measured gene expression of PAR_2_ in intestinal and colon stem cells [43,44]. In murine organoid assays, PAR_2_ activation has been found critical to protect colon progenitors from anoïkis [42]. PAR_2_ activation has induced a decrease in proliferation of epithelial progenitors, by contrast with PAR_1_ [42]. Downstream of PAR_2_ activation, a signaling pathway implicating β arrestin-2, PP2A and GSK3β has been triggered and regulated by the cytoskeleton organizer Rho, pushing colon stem cells towards a quiescent and resistant phenotype. This quiescence could result from GSK3β-dependent β-catenin degradation and/or the increased expression of the DUSP6 phosphatase, a negative regulator of ERK signaling [42]. Interestingly, it has been shown that active GSK3β could positively regulate the phosphatase PP1 [45] which stabilizes the co-transcription factor YAP downstream of PAR_2_ to promote epithelial survival and regeneration in response to injury [46].

Importantly, as for hematopoietic stem cells, a sexual dimorphism in PAR_2_-dependent GSK3β activation in colon stem cells has been observed [47]. The PAR_2_/GSK3β pathway has been triggered in colon progenitors from male mice, whereas females have displayed a PAR_2_/AKT pathway. Moreover, PAR_2_ has been shown to control specific gene expression in males and females, i.e., Itga6 and Timp2, respectively.

Thus, active GSK3β has a protective role for adult stem cells from both mesenchymal and epithelial origin, acting as a sensor of their niche, and under plastic regulation in both sexes.

The Figure 4 summarizes data on PAR_2_-dependent regulation of GSK3β in colon stem cells.

## 6. Consequences of Prolonged GSK3β Activation in Inflammation and Cancer

GSK3β activation has a protective role for adult stem cells through its capacity to switch them towards a quiescent and resistant phenotype. However, in a deregulated microenvironment, prolonged or iterative activation of GSK3β can be deleterious. Factors responsible for deregulated microenvironment include chronic inflammation with high levels of cytokines and proteases, persistent matrix remodeling and metabolic changes. It is thus clear that the restoration of a stem cell niche with physiological functions of integrins and protease-activated receptors is critical to control GSK3β activity.

In inflammation, long-term GSK3β activation should be deleterious through a decrease of regenerative capacities. Indeed, after acute inflammation, stem cells maintain their quiescent phenotype showing decreased proliferation and migration, as well as poor differentiation. In addition to immunity control, tissue regeneration is a major aim in IBD therapy [48].

Prolonged GSK3β activation in inflammatory niche could maintain abnormal stem cells, avoiding apoptosis (i.e., mitotic catastrophe). As a result, stem cells accumulating mutations or chromosomal aberrations reside in the tissue until the release of their dormant state by epigenomic or genomic changes inducing proliferation [4]. These pre-cancerous stem cells are characterized by genomic instability and re-expression of embryonic genes, as well as dependence to the normal niche [49]. Such pre-leukemic stem cells with high GSK3β activity have been described in transition to cancer stem cells with deregulated β-catenin [50]. Importantly, due to the different pools of GSK3β, an aggressive cancer stem cell can cumulate both active GSK3β-dependent survival and β-catenin-dependent self-renewal deregulation. Indeed, we have measured higher clonogenic capacities of leukemic progenitors displaying GSK3β-dependent drug resistance [26]. 

The capacity to activate GSK3β in adult stem cells could be the property of specific receptors after their binding to matrix proteins (α_5_, α_2_ integrins, [25,51]) or their protease-dependent cleavage (PAR_2_, [42]). Also, the GSK3β activation could result from the dialog between integrins and PAR_2_ [42,52]. The α_5_ and α_2_ integrin-binding proteins, fibronectin and collagen respectively, and PAR_2_ have been implicated in cell plasticity [53,54,55]. Prolonged GSK3β activation could induce a switch in stem cell identity that influences the responses to the microenvironment, paving the way to tumor transformation [56] (Figure 5).

## 7. GSK3β as a Target for Precision Medicine

GSK3β has an important theranostic potential in pathologies where stem cells are deregulated, such as inflammation [57] and cancer [58]. It seems that the activated GSK3β signature could involve subsets of pathological progenitor/stem cells depending both on their membrane receptors and the adhesive and protease activities in their niche. For example, GSK3β has a key oncogenic role in leukemia with MLL mutations [50] and intestinal neuroendocrine tumors [59]. Interestingly, the cancer stem cells in those pathologies could be developed from early progenitors [60] through the dialog with an inflammatory [61,62] and neurologic-deficient [63,64] microenvironment.

Clinical parameters as gender, age and metabolic status of patients should strongly influence therapeutic decisions aiming to target GSK3β in inflammatory and cancer diseases. We have seen above that the sexual dimorphism occurring in GSK3β-dependent regulation of adult stem cells could be inversed during the transition from inflammation to cancer. Also, aging and obesity could modify cell responses to insulin and to stresses. Therefore, therapeutic targeting of GSK3β must be thought as a personalized medicine.

Measurement of the activity of GSK3β is complex due to its different cellular pools with independent regulations. However, when human samples are available, 2D or 3D stem cell cultures with controlled addition of growth factors represent good investigating tools for pre-clinical assays with adhesive conditions akin to the tissue architecture [26,42]. Also, active GSK3 imaging agents are in development for positron emission tomography as a diagnostic tool but require yet a sufficient knowledge of pathways governing GSK3β regulation [65].

Natural and synthetic GSK3 inhibitors with different modes of action are already commercialized and some of them are in clinical trials for the treatment of neurodegenerative diseases and cancer [65,66]. We and others have found that natural compounds and their derivatives are potential drugs to kill cancer stem cells through GSK3β inhibition [67,68]. It is important to note that GSK3 inhibitors represent also interesting therapeutic tools to restore normal niche functions [69]. Thus, targeting both stem cells and their niche is crucial to restore physiological tissue regeneration (Figure 6).

## 8. Conclusions and Future Directions

GSK3β is a master kinase in the regulation of adult stem cells through the control of common mechanisms in different cells such as hematopoietic and intestinal stem cells. It is a critical sensor of the microenvironment allowing important phenotypic changes in stem cells for their adaptation and their maintenance. Our previous work demonstrated that the GSK3β activation is supervised by a balanced control between adhesion and protease-activated receptors in the regulation of adult stem cell behavior.

However, the protective role of GSK3β can be perverted in pathogenesis by the maintenance of stem cells with functional deficits and genetic aberrations. This is the reason why the interest for GSK3β as a therapeutic target continues to grow with increased rate of publications and pre-clinical trials. In inflammation and cancer, targeting of GSK3β could restore a normal interaction of stem cells with their microenvironment and consequently homeostatic tissue regeneration. An increasing number of studies now strongly suggests that therapeutic targeting of GSK3β must take into account clinical parameters such as gender and metabolic status of patients.

Progress is required both in fundamental and clinical research to improve our knowledge of GSK3β. As an ancestral kinase of the stress response in eukaryotes, attention should be drawn towards works on its orthologue shaggy in drosophila and other primitive organisms. Indeed, to cope with stress, vertebrate adult stem cells develop mechanisms based on ancient roots. Also, for a theranostic purpose, diagnostic and therapeutic tools specific for GSK3β versus its isoform GSK3α are necessary. Rapid advances in the research about neurodegenerative disorders should offer opportunities in this field.

## Figures and Tables

**Figure 1 cells-10-00225-f001:**
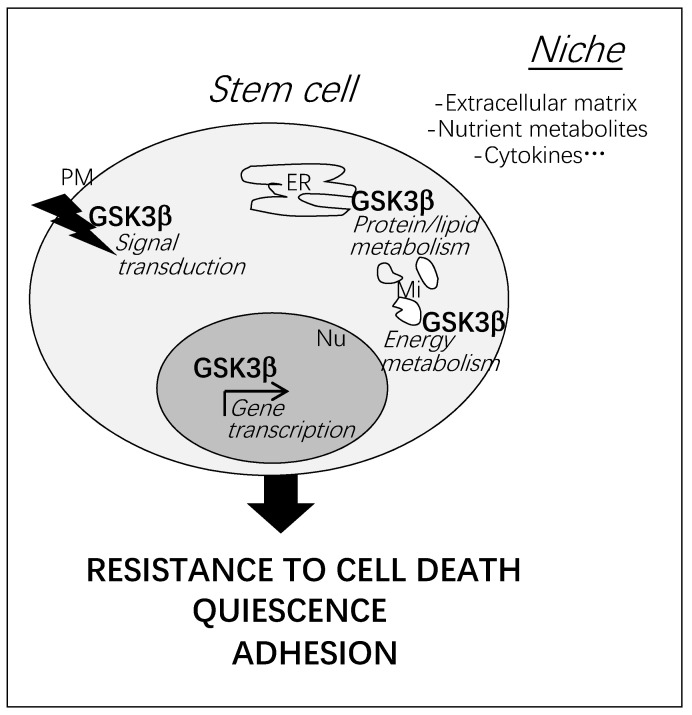
Multiple cellular localization of GSK3β activation and consequences on stem cell status. At the plasma membrane, GSK3β is critical to counteract death-signaling pathways induced by cytokines and adhesion changes in injured microenvironment. In case of nutrient or oxidative stresses, GSK3β can adjust metabolic activities in the endoplasmic reticulum and in mitochondria. Cell survival and metabolism are tightly linked and their cross-regulation is particularly important in subcellular compartments where GSK3β is found. In the nucleus, GSK3β−modulated gene transcription contributes to shape the resistant phenotype of stem cells. Localized signaling activities responsible for GSK3β activation are complex as well as the relationship between them [1]. PM: Plasma membrane; ER: Endoplasmic reticulum; Mi: Mitochondria; Nu: Nucleus.

**Figure 2 cells-10-00225-f002:**
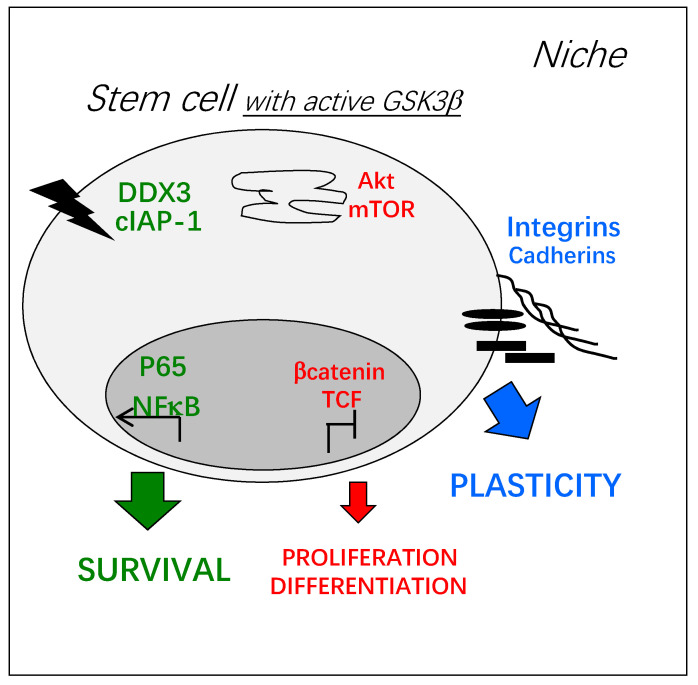
Key cellular functions potentially regulated by active GSK3β in adult stem cells: Survival (green) and plasticity (blue) are promoted and metabolism/proliferation/differentiation (red) are decreased. Depending on the environment, signal transduction at the plasma membrane can trigger GSK3β activation, by the inhibition of its kinase-dependent phosphorylation or the activation of its phosphatase-dependent de-phosphorylation, for example [1]. Inside the cell, active GSK3β can be found in different molecular complexes and its trafficking between them is poorly known. These pools of GSK3β can be independently activated or inhibited and little is known on their coordinated activation.

**Figure 3 cells-10-00225-f003:**
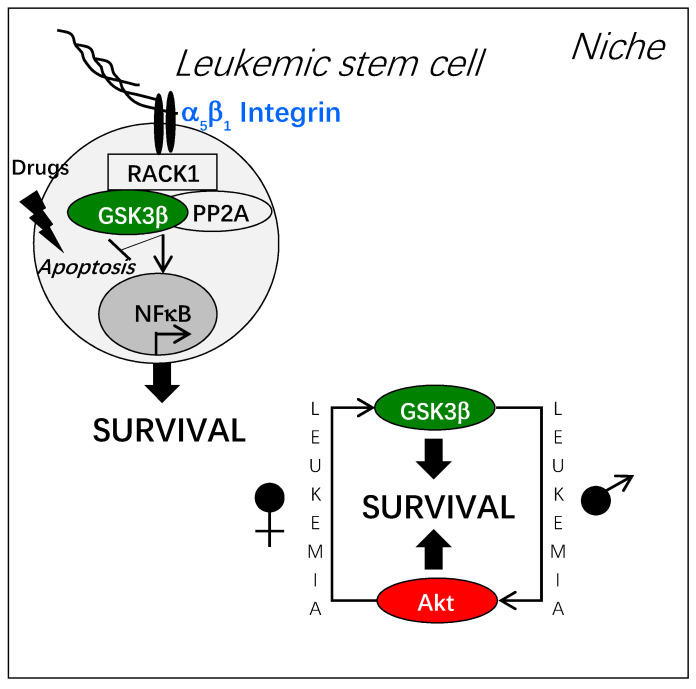
Integrin-dependent activation of GSK3β and leukemic stem cell resistance. Adhesion of leukemic stem cells to fibronectin or to osteoblasts triggers the formation of a molecular complex around the adaptor protein RACK1 associated to the cytoplasmic domain of integrin. In this molecular complex, GSK3β is activated through its de-phosphorylation by the phosphatase PP2A. Thus, leukemic stem cell resistance to cytotoxic stress is promoted by different mechanisms, among them NF-kB activation. Of note, the dependence of hematopoietic stem cells to GSK3β displays a sexual dimorphism that is switched upon leukemogenesis.

**Figure 4 cells-10-00225-f004:**
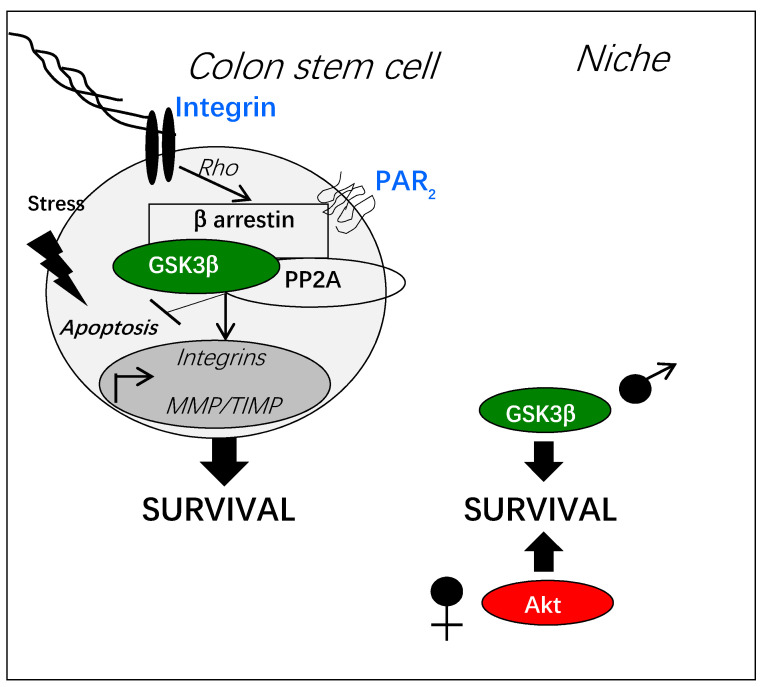
PAR_2_-dependent activation of GSK3β in normal colon stem cells. PAR_2_ activation by protease-dependent cleavage can induce a β arrestin-dependent signaling pathway resulting in GSK3β activation through its de-phosphorylation by the phosphatase PP2A. This signaling pathway is controlled by the Rho kinase activated by adhesive stress. Gene transcription of integrins, metalloproteases (MMP) and their inhibitors (TIMP), is regulated by the PAR_2_/GSK3β pathway. Active GSK3β counteracts stress-induced apoptosis by multiple mechanisms (see Section 3, “GSK3β and key functions of adult stem cells”). In the physiology of colon stem cells, the pro-survival signaling of PAR_2_ is sexually dimorphic depending on GSK3β or on Akt.

**Figure 5 cells-10-00225-f005:**
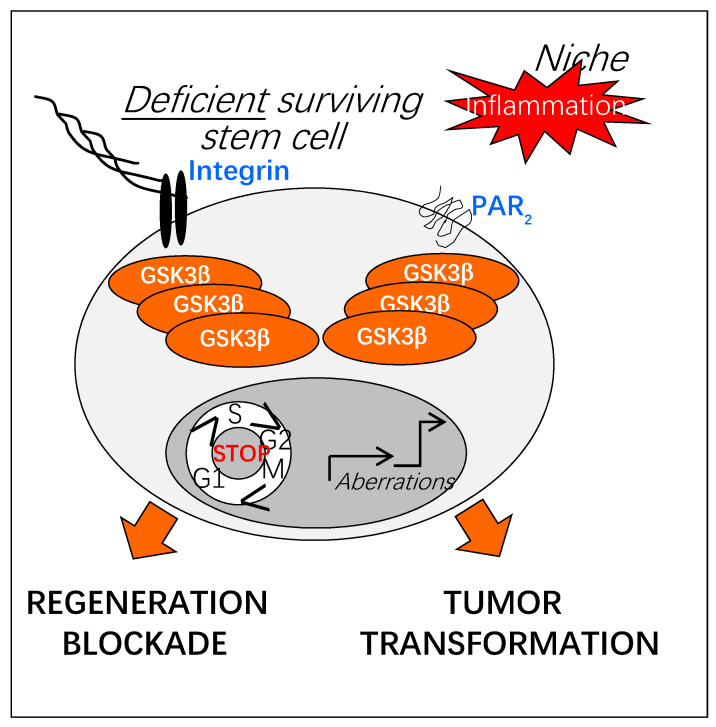
Potential impact of prolonged GSK3β activation on regenerative functions of adult stem cells. Upon chronic inflammation, increased activation of GSK3β could result in a deregulation of the regenerative properties of adult stem cells (defect in tissue repair, tumor transformation). As sensors of the injured stem cell niche, adhesion and protease-activated receptors (Integrins, PARs) could play a crucial role in the sustained activation of GSK3β.

**Figure 6 cells-10-00225-f006:**
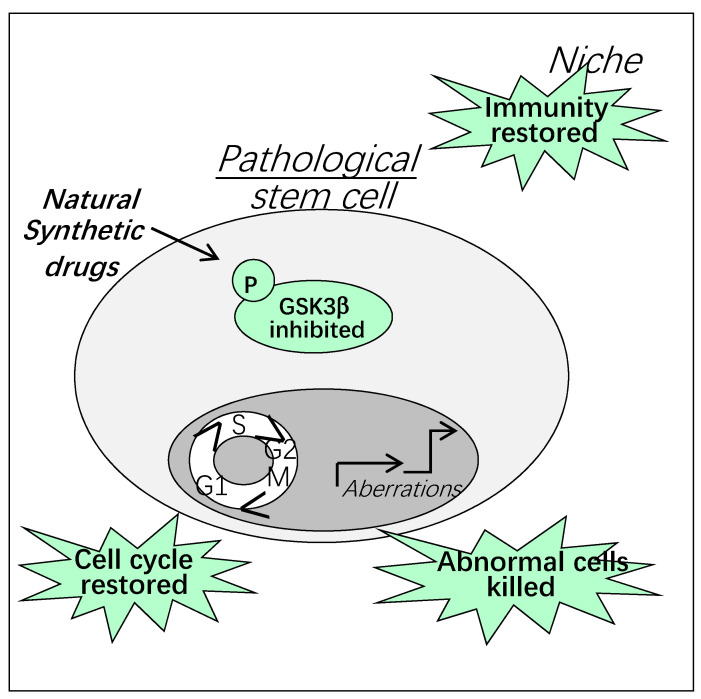
Therapeutic targeting of GSK3β in pathologic stem cells and their niche.

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
