# Peer review of "GSK3β, a Master Kinase in the Regulation of Adult Stem Cell Behavior"

_cells, 2021, doi:10.3390/cells10020225_

Round 1
Reviewer 1 Report
Racaud-Sultan C and Vergnolle N reviewed GSK3β biology and pathology in adult/somatic stem cells by focusing on hematopoietic system and colon crypts as well as on their neoplastic counterparts. This is a timely and important topic in the field of stem cell biology. As commented below, however, the context and the flow of description are unfluent and fragmented to conclude the roles of GSK3β in normal and pathological adult/somatic stem cells. The authors are encouraged to address the following issues.
1. General comments
It is defined in WikipediA (https://en.wikipedia.org/wiki/Adult_stem_cell) that adult stem cells are undifferentiated cells, found throughout the body after development, that multiply by cell division to replenish dying cells and regenerate damaged tissues. It is also stated that scientific interest in adult stem cells is centered on their ability to divide or self-renew indefinitely, and generate all the cell types of the organ from which they originate, potentially regenerating the entire organ from a few cells. Therefore, characteristics and biology of adult/somatic stem cells and their niche environments are specific to the corresponding organs/tissues and are variable and heterogeneous among the different organs and tissues. However, the authors have too simplified and fractionalized the roles of GSK3β in such various somatic stem cells and niche environment. They have also frequently discussed GSK3β biology in normal and pathological somatic stem cells in the same platform. These points should be substantially addressed and improved.
Following the original content of this review manuscript and as commented below, the authors are encouraged to sequentially and systematically review the GSK3β biology in normal hematopoietic and colon stem cells, then the GSK3β pathology in stem/progenitor cells in leukemia and colon cancer, and ultimately the GSK3β- and stem cells-targeted therapy against these cancer types.
2. GSK3β as a sensor of the adult stem cells
This section describes the subcellular localization-dependent roles of GSK3β in adult stem cells in response to the stress and stimulations from niche environment. However, it is unclear that the respective subcellular fractions of GSK3β react to what types of stress/stimulations from their niche, and how niche stresses activate GSK3β at different cell compartments. Figure 1 should also clearly depict the relationship between them. As commented above, is this figure context common to somatic stem cells in all or most organs and tissues? What do the terms “Transduction”, “Synthesis”, “Metabolism”, “Transcription” and “RESISTNCE” in Figure 1 mean?
It is well known that GSK3β plays these pivotal roles also in various differentiated somatic cells in response to intracellular and extracellular stimuli. What are the differences in GSK3β biology between somatic stem cells and differentiated somatic cells in the same cell lineage? The authors are encouraged to discuss to clarify these issues.
3. GSK3β and key functions of adult stem cells
This section describes the involvement of GSK3β in stemness phenotypes of “adult stem cells with active GSK3β”. The points of arguments in this section are inconsistent and unconvincing. This is because the authors discuss the localization-dependent functions of GSK3β based on the previous studies on mouse (normal) hematopoietic stem cells [9], human colon cancer cells [10, 12], HeLa (uterine cervical squamous cell carcinoma), C57MG (human mammary epithelial, probably non-stem) and NIH3T3 (mouse fibroblast, probably non-stem) cells [11], HeLa cells and MDA-MB-231 human breast cancer cells [13], and human chronic lymphocytic leukemia B cells [14]. Ultimately, the authors dictate the roles of GSK3β in adult (probably normal) stem cells by gathering the knowledge reported from normal and neoplastic stem and non-stem cells in the same platform.
Figure 2 shows the dual roles for GSK3β in promoting cell survival and in suppressing proliferation (self-renewal) and differentiation in the adult stem cells with active GSK3β. What mechanism(s) dictates these bidirectional roles of GSK3β in the same adult stem cells?
Do adult stem cells with inactive GSK3β exist? If so, it is important to biologically compare adult stem cells with active and inactive GSK3β.
4. Integrin-dependent activation of GSK3β in leukemia
This section brusquely and superficially describes the pathological roles of GSK3β in leukemia cells and their progenitor cells mainly based on the authors’ previous studies. There are many comparative original studies and reviews on GSK3β biology in normal hematopoietic stem cells (e.g., Trowbridge JJ, et al. Nat Med 2006;12:89-98; Ko KH, et al. Stem Cells 201029:108-18; Lapid K, et al. J Clin Invest 2013;123:1705-17) and in leukemia progenitor/stem cells (e.g., Holmes T, et al. Stem Cells 2008;26:1288-97; Holmes T, et al. Curr Med Chem 2008;15:1493-9; McCubrey JA, et al. Leukemia 2014;28:15-33). These previous studies showed the oposing roles of GSK3β between normal and neoplastic hematopoietic stem cells. Therefore, for better understanding of this section in the whole context of this review, it is important to comprehensively review GSK3β biology in normal and neoplastic hematopoietic stem/progenitor cells.
5. PAR2-dependent activation of GSK3β in the colon crypt
As the authors reviewed in the section 3, GSK3β is a critical member of the regulators of the Wnt/β-catenin pathway that apparently plays a crucial role in regulation of colon crypt stem cells. Therefore, it is suggested to discuss the consequence or influence of PAR2-dependent activation of GSK3β in the Wnt/β-catenin pathway in colon crypt stem cells.
6. Consequences of prolonged GSK3β activation in inflammation and cancer
For better understanding of this section, it is important to discuss biological mechanism(s) for “prolonged GSK3β activation in inflammatory (and probably neoplastic) niche”. Does Figure 5 commonly represent prolonged, active GSK3β both in leukemia (hematopoietic) and colon cancer (crypts) stem cell niches?
Reviewer 2 Report
In this manuscript the authors make a detail review of the literature regarding the role of GSK3β in the regulation of adult stem cell behaviour. This review focus on the activation of GSK3β by integrins and protease activated receptors (PARs) and the implications of this activation in inflammation and cancer. The authors are experts in the field. Overall, I feel that this review is well organized and of interest to the cancer stem cell field. However some sentences should be revised for clarity of the English language (highlighted below). For the reasons stated above it is my opinion that this manuscript should be accepted for publication after minor changes which I describe below.
Minor comments:
I would like to suggest changing: “a sexual dimorphism” to a gender dimorphism. In fact, I think that in the context of this manuscript the word gender (instead of sexual) is more appropriate and this should be checked throughout the manuscript.
The authors should check the manuscript for consistency of verbal tense. This should be uniform throughout the manuscript.
Line 48: where it reads: “plasmatic membrane” it should read: plasma membrane
Lines 50-52: The following sentence needs revision to become clearer: “Thus, a resistant and quiescent stem cell phenotype is established characterized by a strong anchorage to the extracellular matrix or to supporting cells in the niche and by a low energetic metabolism.”
Line 56: I would suggesting changing: “Beside its role in survival of adult stem cells, active GSK3β prevents ROS-induced” to: In addition to its role in survival of adult stem cells, active GSK3β prevents ROS-induced
Figure 1 legend: Where it reads: “Plasmic membrane” it should read: plasma membrane. Please check and replace this throughout the manuscript.
Lines 89-91: The following sentence needs revision for clarity: “An IκB-independent activation of NF-κB is due to the direct regulation of the p65 subunit by GSK3β in the nucleus. By this way, transcriptional targets of NF-κB are specific through a GSK3β-dependent epigenetic control.”
Lines 113-117: This paragraph should be revised for clarity of the English language.
Line 135: Where it reads: “characterized by an high activity”, it should read: characterized by a high activity
Lines 143-145: “We measured the epithelial expression of PAR2 and PAR1, two members of the PAR family implicated in 144 IBD and CRC, along the colon crypt [26].” Are the authors the only group that showed this? Otherwise they need to add references for other studies that have also looked into this.
Lines 168-169: Where it reads: “In therapy of IBD, the release of regeneration is a major aim beside the control of immunity [28].” it should read: In addition to immunity control, the release of regeneration is a major aim in IBD therapy [28].
Lines 180-185: This paragraph should be revised for clarity of the English language.
Line 223: Where it reads: “GSK3β is definitively a master kinase” please replace with: GSK3β is a master kinase
Line 226: Where it reads: “Our work demonstrates”, it should read: Our previous work demonstrated. I would also like to know if this has been shown exclusively by authors or if others have also shown this. This is a review article and as such the authors need to make sure to acknowledge all available literature.
Line 229: “However the protective role of GSK3β can be devoured in pathogenesis”. I am confused as to the meaning of the word devoured in this context. Could the authors clarify or replace by a clearer word?
Line 230: Where it reads: “It is why”, it should read: This is the reason why
Round 2
Reviewer 1 Report
As described below, the authors revisions and responses to the comments raised at the 1st round of review are still insufficient to improve this manuscript. Additional comments are provided here based on the point-by-point responses to the comments at the 1st round of review.
- General comments
It is defined in WikipediA (https://en.wikipedia.org/wiki/Adult_stem_cell) that adult stem cells are undifferentiated cells, found throughout the body after development, that multiply by cell division to replenish dying cells and regenerate damaged tissues. It is also stated that scientific interest in adult stem cells is centered on their ability to divide or self-renew indefinitely, and generate all the cell types of the organ from which they originate, potentially regenerating the entire organ from a few cells. Therefore, characteristics and biology of adult/somatic stem cells and their niche environments are specific to the corresponding organs/tissues and are variable and heterogeneous among the different organs and tissues. However, the authors have too simplified and fractionalized the roles of GSK3β in such various somatic stem cells and niche environment. They have also frequently discussed GSK3β biology in normal and pathological somatic stem cells in the same platform. These points should be substantially addressed and improved.
We understand the remarks of the Referee and we have tried to improve the clarity of the text. However, we would underline that our objective was to write a review pointing only some aspects of the role of GSK3β in stem cells. Indeed, we answered to a special issue proposed by CELLS and we thought that general considerations on the roles of GSK3β could be found elsewhere. Our objective was to highlight common roles played by GSK3β in both normal and pathological somatic stem cells, through the examples of leukemic and colon stem cells. We believe that this approach is important to understand the fundamental role of GSK3β in adult stem cells and its importance in pathogenesis. We hope that our manuscript will offer a new point of view to the scientific community since the number of reviews on GSK3β is already plethoric.
Following the original content of this review manuscript and as commented below, the authors are encouraged to sequentially and systematically review the GSK3β biology in normal hematopoietic and colon stem cells, then the GSK3β pathology in stem/progenitor cells in leukemia and colon cancer, and ultimately the GSK3β- and stem cells-targeted therapy against these cancer types.
As requested by the Referee and indicated in the different sections below, we have completed information relative to hematopoietic and colon stem cells, in normal and pathological conditions.
[Comments at the 2nd round of review]
If the authors focus on the role of GSK3 in hematopoietic and colon stem cells in normal and pathological conditions, but not aim to comprehensively review its role in adult stem cells in general, the Title and the introductory part of Abstract should mislead readers including Reviewer 1. “Hematopoietic and colon stem cells in normal and pathological conditions” should be specified in Title and in the beginning of Abstract. This also should be specified in Introduction. Accordingly, “adult stem cells” and “hematopoietic and colon stem cells” should be adequately distinguished throughout the manuscript to prevent misleading and confusion.
- GSK3β as a sensor of the adult stem cells
This section describes the subcellular localization-dependent roles of GSK3β in adult stem cells in response to the stress and stimulations from niche environment. However, it is unclear that the respective subcellular fractions of GSK3β react to what types of stress/stimulations from their niche, and how niche stresses activate GSK3β at different cell compartments. Figure 1 should also clearly depict the relationship between them. As commented above, is this figure context common to somatic stem cells in all or most organs and tissues? What do the terms “Transduction”, “Synthesis”, “Metabolism”, “Transcription” and “RESISTNCE” in Figure 1 mean?
As requested by the Referee, we have clarified the Figure 1 by addition of precisions to the terms employed and explanations in the legend. Concerning the regulation of the activation of GSK3β, given the complexity of that regulation with various modes (serine-dephosphorylation, tyrosine phosphorylation, dimerization, substrate priming…), we believe that is out of the scope of our review. Our objective was to only point out the subcellular compartments where GSK3β plays a critical role in response to microenvironment changes. Moreover, little is known in the relationship between subcellular fractions for the GSK3β regulation. Nutrient and oxidative stresses, as well as injury by inflammatory cytokines, are common to all types of adult stem cells and activities of the ancestral kinase GSK3β are fundamental to offer protection even in primitive organisms.
[Comments at the 2nd round of review]
As commented above, it is critical to clarify whether this section and Figure 1 focus on “general adult stem cells” or “hematopoietic and colon stem cells”. It is also better to depict the stress/stimuli from niche in Figure 1 according to its legends. This is not so difficult.
It is well known that GSK3β plays these pivotal roles also in various differentiated somatic cells in response to intracellular and extracellular stimuli. What are the differences in GSK3β biology between somatic stem cells and differentiated somatic cells in the same cell lineage? The authors are encouraged to discuss to clarify these issues.
We agree with the remark of the Referee given the survival role played by the kinase GSK3β in resting cells such as germinal center B cells [Jellusova et al., Nat Immunol, 2017, 18, 303]. However, the specific role of GSK3β in stem cells and progenitors is to balance survival/metabolism pathways with self-renewal/differentiation pathways. Indeed, GSK3β is central to regulate morphogenetic pathways such as Wnt, Notch, Shh and thus controls stem cell behavior. Specific cues in the microenvironments could explain why GSK3β modulation can impact stem cells without affecting mature cells of the lineage. We have now included this point in the section « GSK3β and key functions of adult stem cells », lines 86-88.
- GSK3β and key functions of adult stem cells
This section describes the involvement of GSK3β in stemness phenotypes of “adult stem cells with active GSK3β”. The points of arguments in this section are inconsistent and unconvincing. This is because the authors discuss the localization-dependent functions of GSK3β based on the previous studies on mouse (normal) hematopoietic stem cells [9], human colon cancer cells [10, 12], HeLa (uterine cervical squamous cell carcinoma), C57MG (human mammary epithelial, probably non-stem) and NIH3T3 (mouse fibroblast, probably non-stem) cells [11], HeLa cells and MDA-MB-231 human breast cancer cells [13], and human chronic lymphocytic leukemia B cells [14]. Ultimately, the authors dictate the roles of GSK3β in adult (probably normal) stem cells by gathering the knowledge reported from normal and neoplastic stem and non-stem cells in the same platform.
We understand the remark of the Referee and regret the confusion given by this paragraph. Actually, fundamental research on the GSK3β roles in various cell types (stem cells or differentiated cells, normal or cancer cells) is of importance given the ubiquitous expression of this kinase. As pointed above by the Referee, GSK3β can have similar functions in differentiated cells such as lymphocytes and stem cells, depending on the microenvironment context (low glucose environment, for example). Moreover, cancer cells re-acquire stem cell properties and thereby their GSK3 regulation can reflect what happens in stem cells. However, following the recommendation of the Referee, we have added information on the potential differences in GSK3β regulation between stem cells and differentiated cells in the same lineage (lines 86-88). Also, we have changed the sentence (line 101) above the Figure 2 for “All these GSK3β-dependent functions could be critical in stem cells ».
[Comments at the 2nd round of review]
This response and the minor revisions in this section do not cover the comments at the 1st-round of review. Ref. No. 17 is for hepatocytes, but not for liver stem cells, and Ref. No. 19 is for leukemia cells, but not for leukemia stem cells. Unless major revisions, the authors are suggested/encouraged to delete whole this section and Figure 2.
Figure 2 shows the dual roles for GSK3β in promoting cell survival and in suppressing proliferation (self-renewal) and differentiation in the adult stem cells with active GSK3β. What mechanism(s) dictates these bidirectional roles of GSK3β in the same adult stem cells?
As now mentioned in the legend of the Figure 2, GSK3β-dependent functions in stem cells depend on different pools of the kinase, independently activated or inhibited. For example, GSK3β can be activated in a complex (DDX3, cIAP-1) associated with death receptors to promote cell survival and in the same cell, can be activated or inhibited in the β catenin-destruction complex, promoting quiescence or proliferation. The mechanisms coordinating these different GSK3β pools are unknown and the potential trafficking of GSK3β between different subcellular compartments is poorly understood. As a result of the microenvironment context, early signal transduction at the plasma membrane may dictate downstream regulation of GSK3β through kinases such as Akt and phosphatases such as PP2A.
[Comments at the 2nd round of review]
According to this response, little literature background is available to support Figure 2 showing the triple roles of GSK3b in somatic (hematopoietic or colon?) stem cells in response to or under the control by stem cell niche.
Do adult stem cells with inactive GSK3β exist? If so, it is important to biologically compare adult stem cells with active and inactive GSK3β.
Depending on the environment, GSK3β can be found active or inactive in different molecular complexes, in all somatic stem cells. Our review focus on the roles of pools with active GSK3β. However, we would underline that neural stem cells represent a model where GSK3β must be very tightly regulated. It is now pointed in the text (lines 102-104). Indeed, it has been shown that upon insulin withdrawal, active GSK3β can trigger cell death by autophagy in neural stem cells [Ha et al., Mol Brain, 2015, 8, 30]. It appears that low GSK3 activity is necessary for optimal survival and proliferation of neural progenitors whereas higher GSK3 activity is required for efficient function of differentiated neurons [Cole, Frontiers in Molecular Neurosciences, 2012, 5]. The niche of neural stem cells could also be largely impaired by an over-activity of GSK3β [Fuster-Mantazo et al., Human Mol Genetics, 2013, 22, 1300].
[Comments at the 2nd round of review]
According to this response, it is necessary to describe this review does not focus on the somatic stem cells with inactive GSK3b. Again, this revision does not always meet the scope or context of this review focusing on hematopoietic and colon stem cells as the authors mentioned above.
- Integrin-dependent activation of GSK3β in leukemia
This section brusquely and superficially describes the pathological roles of GSK3β in leukemia cells and their progenitor cells mainly based on the authors’ previous studies. There are many comparative original studies and reviews on GSK3β biology in normal hematopoietic stem cells (e.g., Trowbridge JJ, et al. Nat Med 2006;12:89-98; Ko KH, et al. Stem Cells 2010;29:108-18; Lapid K, et al. J Clin Invest 2013;123:1705-17) and in leukemia progenitor/stem cells (e.g., Holmes T, et al. Stem Cells 2008;26:1288-97; Holmes T, et al. Curr Med Chem 2008;15:1493-9; McCubrey JA, et al. Leukemia 2014;28:15-33). These previous studies showed the opposing roles of GSK3β between normal and neoplastic hematopoietic stem cells. Therefore, for better understanding of this section in the whole context of this review, it is important to comprehensively review GSK3β biology in normal and neoplastic hematopoietic stem/progenitor cells.
Following the recommendation of the Referee, we have now added information on hematopoietic stem cells and leukemic stem cells in this section (lines 118-131).
[Comments at the 2nd round of review]
This section has been substantially revised, and thus covers the comment at the 1st round of review.
- PAR2-dependent activation of GSK3β in the colon crypt
As the authors reviewed in the section 3, GSK3β is a critical member of the regulators of the Wnt/β-catenin pathway that apparently plays a crucial role in regulation of colon crypt stem cells. Therefore, it is suggested to discuss the consequence or influence of PAR2-dependent activation of GSK3β in the Wnt/β-catenin pathway in colon crypt stem cells.
Following the “General Comments” of the Referee and in response to his remark on PAR2-dependent activation of GSK3β and its impact on Wnt signaling, we have added information on the roles of active GSK3β in colorectal cancer cells (lines 166-175) and on the mechanisms of quiescence downstream of PAR2/GSK3β signaling (lines 187-191).
[Comments at the 2nd round of review]
This section has been substantially revised, but it needs to clarify whether Figure 4 represents normal colon stem cells or colon cancer stem cells.
- Consequences of prolonged GSK3β activation in inflammation and cancer
For better understanding of this section, it is important to discuss biological mechanism(s) for “prolonged GSK3β activation in inflammatory (and probably neoplastic) niche”. Does Figure 5 commonly represent prolonged, active GSK3β both in leukemia (hematopoietic) and colon cancer (crypts) stem cell niches?
As requested by the Referee, we have now added examples of deregulated microenvironment responsible for prolonged GSK3β activation (lines 213-219). The Figure 5 aims to depict the potential impact of a sustained activation of GSK3β on the regenerative capacities of adult stem cells, in general. Also is highlighted the importance of both adhesion receptors and protease-activated receptors in this over-activation of GSK3β. Explanations are now given in the Figure 5 legend.
[Comments at the 2nd round of review]
According to the revised description in this section, Figure 5 appears to depict the roles of GSK3b in normal colon stem cells, but not in the adult stem cells “in general”.
